# A Fiber-Optic Sensor-Embedded and Machine Learning Assisted Smart Helmet for Multi-Variable Blunt Force Impact Sensing in Real Time

**DOI:** 10.3390/bios12121159

**Published:** 2022-12-13

**Authors:** Yiyang Zhuang, Taihao Han, Qingbo Yang, Ryan O’Malley, Aditya Kumar, Rex E. Gerald, Jie Huang

**Affiliations:** 1Research Center for Optical Fiber Sensing, Zhejiang Laboratory, Hangzhou 311121, China; 2Department of Electrical and Computer Engineering, Missouri University of Science and Technology, Rolla, MO 65409, USA; 3Department of Materials Science and Engineering, Missouri University of Science and Technology, Rolla, MO 65409, USA; 4Cooperative Research, College of Agriculture, Environmental and Human Sciences, Lincoln University of Missouri, Jefferson City, MO 65102, USA

**Keywords:** mild traumatic brain injury, fiber-optic sensor, fiber Bragg grating, machine learning, bunt force impact

## Abstract

Early on-site diagnosis of mild traumatic brain injury (mTBI) will provide the best guidance for clinical practice. However, existing methods and sensors cannot provide sufficiently detailed physical information related to the blunt force impact. In the present work, a smart helmet with a single embedded fiber Bragg grating (FBG) sensor is developed, which can monitor complex blunt force impact events in real time under both wired and wireless modes. The transient oscillatory signal “fingerprint” can specifically reflect the impact-caused physical deformation of the local helmet structure. By combination with machine learning algorithms, the unknown transient impact can be recognized quickly and accurately in terms of impact magnitude, direction, and latitude. Optimization of the training dataset was also validated, and the boosted ML models, such as the S-SVM+ and S-IBK+, are able to predict accurately with complex databases. Thus, the ML-FBG smart helmet system developed by this work may become a crucial intervention alternative during a traumatic brain injury event.

## 1. Introduction

Mild traumatic brain injuries (mTBIs), which are resulted from mechanical energy to the head from external forces, frequently occur among sports and military personnel [1,2]. mTBIs can cause multiple temporary or permanent physical and mental diseases such as post-traumatic stress disorder, amnesia, depression, persistent headache, etc. Compared with acute TBIs, mTBIs are cumulative and difficult to diagnose at the time of trauma. Since improved outcomes are reported for patients that receive treatments within 60 min of injury, it is significant to identify the mTBIs within the golden hour [3,4]. Therefore, there is a critical need to develop a technology that enables the autonomous collection of trauma-inducing data, as well as subsequent processing reliably and intelligently for prompt TBI identification.

In recent years, efforts have been devoted to developing wearable mTBI prediction devices. Among these devices, mTBI predictions are mostly achieved by measuring the acceleration that the brain suffers using accelerometers [5,6,7,8,9]. Accelerometer-based wearable devices are practical in moving scenarios since the accelerometers only need to be fixed on the wearable devices to measure brain movement. Meanwhile, the accelerometers can be easily integrated with wireless communication modules, thus enabling remote sensing of concussive events for further injury treatment. However, these devices can only provide information about acceleration and magnitude (against a population-based threshold) and almost nothing regarding the impact location, impact severity, or risk of a potential concussion. An alternative method for mTBI prediction is to measure the impact events directly on the head with a strain or vibration sensor embedded in the protective gear. Compared with the accelerometer-based mTBI measurement devices, multiple strain or vibration sensors embedded in a helmet can not only locate the impact events but also estimate the impact forces.

The fiber optic sensor is a promising solution for such strain and vibration measurement. Owing to its unique advantages, such as light weight, small size, high flexibility, etc., the fiber optic sensor can be embedded into the helmet without influencing the functionality of the structure [10,11]. As one of the widely applied fiber optic sensors, the fiber Bragg grating (FBG) has attracted many interests for sensing applications such as strain [12,13], vibration [14,15], and impact [16,17,18]. Compared with other types of widely used single-point fiber optic sensors, such as fiber optic interferometers [19,20,21,22] and long-period fiber gratings [23,24], FBG has a simple and robust in-line structure. Also, multiple FBGs can be cascaded in one optical fiber, and their signals can be individually interrogated by wavelength division multiplexing. In addition, the Bragg wavelength shift can be demodulated by simple algorithms where the shift is proportional to the strain/temperature changes that are applied to the FBGs. Tsuda et al. developed an FBG vibration sensing system employing a fiber ring laser, which can achieve high sensitivity for vibrations ranging from a few hertz to ultrasound frequencies [15]. Kirikera et al. developed a structural health monitoring system based on a network of FBGs, which acquired the impact signals and located the impact events at frequencies up to 180 kHz [17]. Butz and Dennison developed an in-fiber grating impact force transducer, which allowed the overall sensor system to capture all relevant spectral components of force transients from impacts for studying head-helmet mechanical interactions from head impacts. The transducer showed good repeatability in validation experiments [18].

However, despite the impact sensing capabilities of the FBG sensors, in the practical scenarios, there exists complexity in analyzing the recorded transient impact data since it may vary with different factors such as location, direction, magnitude, and impactor type (implementation). A promising solution to enhance the analytical capabilities for the multifaceted analysis of complex transient signals is the machine learning (ML) method. ML algorithms have been introduced to many areas of fundamental and applied research to extract complicated and important attributes from high-value data. Recently, ML has also drawn considerable attention in the field of optical fibers. Liehr et al. have proposed an optical fiber real-time dynamic strain sensing system using the artificial neural network (ANN), in which the ANN significantly reduced the data analysis time and outperformed the standard correlation algorithms [25]. Thompson et al. proposed an ML-calibrated micro-scale fiber-optic force sensor, in which both singular value decomposition-regression analysis and ANN were applied to obtain reliable and accurate force predictions [26]. Wu et al. reported a Brillouin optical time-domain analyzer assisted by the support vector machine (SVM) for ultrafast temperature extraction, which can process about 100,000 Brillouin spectra along 40 km at 250 M samples per second sampling rate for 16 s [27]. Makarenko et al. also presented an approach using ML algorithms for signal recognition in long perimeter monitoring distributed optical fiber sensors, which can recognize seven classes of signals along a 50 km pipeline [28]. These ML algorithms, involving fiber optic sensing applications, indicated that the combination of fiber-optic principles with carefully selected ML models could not only reduce the complexity of data analysis but also enhance the sensing performance of the fiber optic sensors.

In this paper, we proposed an FBG-equipped smart helmet for real-time multimetric blunt force impact sensing via trained ML models. A femtosecond laser-fabricated FBG sensor was embedded into a groove on the outer surface of a football helmet. A pendulum impact system (PIS) was designed and built in-house to apply a series of sophisticated impacts on the football helmet. The transient oscillatory FBG signals were captured in real time and saved for ML training and prediction. Several machine learning (ML) models were developed and implemented in data processing and impact feature recognition. Skipping all the tedious steps of manual computation with the usage of ML models may significantly enhance the overall efficiency and accuracy of the data analysis. Finally, a group of different ML models was developed and optimized for the accurate prediction of varied impact parameters.

## 2. Experimental

### 2.1. The Working Principle and Fabrication of Fiberoptic FBG Sensors

The FBG is formed by creating a periodic refractive index variation in the core of an optical fiber. Such periodic pattern functions as a notch filter that reflects the light of a particular wavelength and transmits others. The pattern can be created using different methods, such as interference lithography, phase mask writing, and point-by-point writing.

The wavelength of the reflected light (*λ_B_*), which is called the Bragg wavelength, is defined as follows:(1)$$\lambda_{B}=2n_{eff}\Lambda$$
where *n_eff_* is the effective refractive index in the core of the optical fiber, and Λ is the grating period. When the FBG sensor is under an applied strain or temperature change, the refractive index in the core of the fiber and the grating period will experience a variation, hence resulting in a shift of Bragg wavelength. Assuming the applied strain or temperature change on the FBG sensor is uniform, the Bragg wavelength shift Δ*λ_B_* introduced by the strain or temperature changes can be described as:(2)$$\frac{\Delta \lambda_{B}}{\lambda_{B}}=(1-p_{e})\Delta \varepsilon +(a_{cte}+a_{cto})\Delta T$$
where *p_e_, a_cte_*, and *a_cto_* are the strain-optic coefficient, thermal expansion coefficient and thermo-optic coefficient, respectively. In practice, since the impact events elapse within several microseconds, the environmental temperature of the FBG sensor can be considered constant. The shift of the FBG wavelength introduced by the axial strain variation can be described as [29]:(3)$$\frac{\Delta \lambda_{B}}{\lambda_{B}}=0.79\Delta \varepsilon$$

Therefore, by continuously monitoring the wavelength of reflected light, the Bragg wavelength shift can be determined, and the changes in the strain applied on the FBG sensor can be calculated. Such changes in strain can cause the Bragg wavelength to be expanded or compressed, depending on the physical deformation of the fiber section containing FBG induced by the impact force (Figure 1a). For instance, when the impact force comes from a perpendicular direction to the FBG axis, the FBG-located fiber section will be forced to expand in the longitudinal direction, hence causing the elongation of the FBG grating period, and resulting in an increased Bragg wavelength (+λ shift). Similarly, when the impact force comes from the longitudinal direction of the FBG, the FBG grating period is thus shortened, and the Bragg wavelength decreases (−λ shift).

To fabricate the fiber optic FBG sensor, a femtosecond laser microfabrication system (Newport) was used (Appendix A). Briefly, four steps were conducted to complete the fabrication process, including (a) sample installation, (b) optical fiber alignment and focal plane adjustment, (c) parameter designation, and (d) linear single-line raster. Detailed steps can be found in the Appendix A.

### 2.2. Fabrication of the FBG-Embedded Smart Football Helmet

A prototype smart helmet based on the FBG sensor is illustrated in Appendix A. A single optical fiber containing one fabricated FBG sensor was embedded in a groove on the surface of the helmet. The groove was located asymmetrically on the top surface of a football helmet and was made with a 1 mm depth and 25 cm length using a heating wire. A piece of optical fiber with an FBG sensor was placed inside the groove and was fixed with epoxy. After embedding, the tail of the optical fiber was spliced into the measurement apparatus for the test. Two types of interrogation methods, a wired and a wireless mode, were respectively used in this study. We initially embedded multiple (10) FBGs into the helmet, and it did increase the quality of the results. However, it would be advantageous if we could measure multiple impact-related parameters (e.g., blunt force impact magnitudes, directions, etc.) with the minimum number of FBGs (e.g., 1 or 2), in which case, the quantity of data processing can be significantly decreased. Meanwhile, we can also cascade other types of fiber optic sensors (e.g., in-line EFPI) in line with FBGs for other applications like low-pressure blast wave monitoring [30,31] and chemical threat detection [32,33,34]. Importantly, an expensive broadband light source and a spectrometer will be required to implement multiplexed FBGs, which would be very challenging for implementing a commercial portable setup where a Wi-Fi-based miniaturized interrogator powered by a button battery may be used for smart helmet applications. Our current wireless interrogator mounted on the helmet is based on an arrayed waveguide gratings (AWGs)-based integrated circuit, which can only interrogate up to 2 FBGs with 1 pre-defined center wavelength.

### 2.3. Signal Interrogation Using Wired and Wireless Methods

Figure 1b shows the general light pathway and interrogation configuration. Under the wired mode, the incident broadband light from the light source (Thorlabs ASE-FL7002-C4 ASE Source) was directed into the FBG sensor embedded on the helmet through an optical fiber circulator (from Port 1 to Port 2). Part of the incident light, which satisfied the Bragg wavelength condition in Equation (1), was reflected and entered the interrogator (BaySpec FBGA-F-1510-1590-FA) via the circulator (from Port 2 to Port 3). Similarly, under the wireless mode (Appendix A), To realize the real-time detection and signal reconstruction, the acquisition speed of the interrogator was set to 5000 Hz, corresponding to a time interval of 0.2 ms.

### 2.4. Establishment of a Pendulum Impact System

To validate the workability of the FBG-embedded smart helmet, a pendulum impact system (PIS) was designed and built in-house to create controlled and repeatable blunt force impact events on a dummy head wearing the helmet. Figure 1c shows a schematic view of the experimental setup. The PIS consists of a metal frame and a bowling ball hung in front of the dummy head via an alloy cable with adjustable length. A dummy wearing the prototype smart helmet was placed in the metal frame. The impact is generated by lifting the bowling ball to a certain height and release to hit the helmet-wearing dummy with zero initial velocity. The location of the manikin was set to ensure a perpendicular impact happened at the lowest spot of the pendulum track. The impact event was monitored and recorded in real time by the interrogator. Impact location on the helmet was controlled to cover a representative, three-dimensional surface area by rotating the manikin 360° with 30° intervals (longitudinal) and using four different bowling ball hanging heights with a step size of 1.5 cm (latitudinal). In addition, at each fixed rotation direction and cable length, the bowling ball was released within a range of kinetic energy by adjusting the ball release height between 25 cm and 55 cm with an interval of 5 cm. Each impact was repeated three times for statistical consideration. Several thousands of impact-induced transient oscillatory signals were acquired and used for the ML model’s training and performance.

### 2.5. Training and Performing of ML Models Using the Collected FBG Datasets

Several standalone and ensemble ML models, including SVM, Random Forest (RF), Multilayer Perceptron-Artificial Neural Network (MLP-ANN), K-nearest Neighbor Instance-Based Learner (IBK), etc., were developed and implemented in this study. The purpose of using ML models is to bypass the classical analysis process, which requires several time-consuming and cumbersome steps, including model development, calibration, and validation. We believe that the ML models—when fully trained—will be able to analyze the transient oscillatory signal datasets and make predictions with superior reliability and accuracy compared to classical models. Furthermore, the ML platform is expected to significantly enhance the ability to identify and isolate patterns in the sensor datasets and ultimately provide a means to predict transient impact-induced concussive events. The ML models used here have either been previously used or considered by researchers working in the field of sensors. A specific introduction to each model can be found in the Appendix A.

All ML models were trained using 75% (750 data records) of the collected transient signal datasets, and their prediction performances were evaluated against the remaining 25% (250 data records) of the sensor signal datasets. The 75% and 25% datasets for training and testing, respectively, were randomly chosen from the data pool of the acquired FBG transient signals obtained from the interrogator. The randomized selection of data records for training and testing is vital for an accurate representation of the parent database—and all of its intrinsic characteristics— in both sets. The 7–25% split in the parent database was chosen because previous studies [17,18,19] reported that such a split is able to ensure proper training of the ML models. The data split is also useful for testing the prediction accuracy of the models in a rigorous manner. Once predictions (of the 25% of the parent database) from the ML models were obtained, they were plotted against the actual values; boundaries of ±10% variations were sketched in the plots to illustrate the accuracy of the predictions.

Five statistical parameters, including person correlation coefficient (R), coefficient of determination (R^2^), mean absolute percentage error (MAPE), mean absolute error (MAE), and root mean squared error (RMSE), were used to evaluate the performance of each ML model. Finally, a single comprehensive parameter was designated as the composite performance index (CPI, see Equation (1)), deriving from the integration of the above five statistical parameters together [18,20].
(4)$$CPI=\frac{1}{N}\sum_{j=1}^{N}\frac{P_{j}-P_{\min,j}}{P_{\max,j}-P_{\min,j}}$$

In Equation (1): N is the total number of performance measures (=5, as five statistical parameters were used in this study); *Pj* is the value of the *j*th statistical parameter; and *P_j,min_* and *P_j,max_* are the minimum (i.e., worst) and maximum (i.e., best) values of the *j*th statistical parameter across the five values generated by the same number of ML models. Based on the formulation shown in Equation (1), the values of CPI would range from 0 to 1, wherein 0 (or the lowest value) would represent the best ML model and 1 (or the maximum value) would represent the worst ML model in terms of overall prediction performance.

## 3. Results and Discussion

### 3.1. Linear Correlation of the Kinetic Energy of the Blunt Force Impact with FBG Signal Wavelength Shift

To evaluate the applicability and robustness of the fabricated smart helmet under the blunt force impact events, a series of simulated impacts using the PIS was conducted using a range of kinetic energy levels from 1.80 J to 19.84 J. Figure 2 exemplifies the results collected at the impact direction of 210°. Each line represents an average level of at least three repeated experiments. We noticed that the amplitude of the resulted signal peaks and valleys might be highly correlated with the magnitudes of the applied kinetic energy. Here, we took the first peak (Figure 2a, red arrow) as the most representative signal and plotted its relative wavelength shift vs. the applied kinetic energy (Figure 2b). A good linear correlation with a coefficient of determination of 0.9781 was found, and a potentially wider linear correlation range may also be testified. The estimated impact speed under the current experimental setup is around 0.89–2.94 m/s (or 3.2–10.6 km/h), which well covers a routine range of low-speed collisions that may occur to the human head. This is also a range where many accumulative mTBIs may occur.

### 3.2. Recognition of Typical “Fingerprint” Features in Raw Transient Oscillatory FBG Signals

Four exemplary plots are shown in Figure 3, representing the first 100 ms transient oscillatory signals resulting from varying the direction and magnitude of the blunt force impacts applied to the smart helmet. All plots show a series of distinguishable spectral features of peaks and valleys. The peaks and valleys that appeared in the early stage after impact (black arrows) are characteristic features of the wavelength shift versus time transients. Under a fixed direction, varied impact kinetic energies caused varied magnitudes of the repeated patterns of peaks and valleys. Changed magnitudes of the signals, again, confirmed the linear correlation of the impact kinetic energy with the signal amplitude, as exemplified in Figure 2.

Meanwhile, different patterns of peaks and valleys were observed when the impact direction varied. These distinguishable patterns can be used as “fingerprint” features to correlate with a unique set of parameters during a blunt force impact event. In the plots, the *x*-axis coordinates of late-stage peaks and valleys were found to shift under different impact energies. Such interfered patterns may be due to complicated internal collisions between the helmet and the manikin head, which are, presumably, less relevant to the initial impact. Thus, the earlier stage (first 30–50 ms) signal “fingerprints” may better reflect the initial impact conditions with higher accuracy. Through such pre-screening of the raw FBG data, it is expected to identify useful signal features and establish analysis categories for the following ML training and performance.

### 3.3. Physical Interpretation Based on the Comprehensive Impact Signals

Figure 4 shows a stacked plot that resulted from a comprehensive test around the smart helmet prototype. Three specific circumstances were also exemplified to introduce the physical insights regarding FBG deformations. The stacked plot includes a total of 28 different impact directions (from 0° to 350°). An expanded view of the transient oscillatory signals within the first 30 ms is also shown to the right (red box). Each spectrum represents an averaged result from three repeated experiments (n = 3), and they were all obtained using the highest impact energy level (19.84 J).

A series of gradually shifted waves of peaks and valleys are observed in Figure 4a. The expanded view of the first 30 ms (right) explicitly shows the right-shifting of all the major peaks (orange bands) and valleys (blue bands) from the top to the bottom of the graph. The “tide-like” trend illustrated here supports the “fingerprint” feature discussed earlier. Similarly, such signal rhythms are mostly accurate within the first 30–40 ms period. After this early stage, fewer and less apparent signal patterns could be identified. We hypothesize these signal variations were reflections of the secondary or tertiary internal collisions after the primary external impact. These patterns enriched the idea of signal “fingerprints” with compelling evidence from 28 different impact directions. In addition, the “tide-like” peaks and valleys evolve with the impact direction, which increases the signal specificity for each impact direction, and will benefit the subsequent training and performance of the ML models.

Three spectrum lines indicated by black arrows, corresponding to the impact directions of 75°, 165° and 260°, were observed with weakened peaks and valleys. Such weakening of signal prominence may be inevitable at the peak/valley inflection points but may not compromise the efficacy of the sensing system under realistic circumstances because (1) it also serves as a unique type of “fingerprint” without peaks or valleys, and (2) its negative influence, if any, can be mitigated by sensor compensation or redundancy in the near future, such as using multiple FBG sensors.

Figure 4b illustrates three speculated circumstances at different impact directions. We hypothesize that the asymmetric location of the FBG sensor on the helmet surface enabled varied physical responses to the helmet deformations. For instance, when impacted from the 0° direction, a force applied longitudinally to the FBG caused a compression effect on the FBG fiber section and shortened the grating period, thus inducing a “valley” in the signal. On the contrary, a force applied perpendicularly to the FBG, as demonstrated in the 260° impact direction, caused an expansion effect on the FBG fiber section, which elongated the grating period, thus inducing a “peak” in the signal. However, when the impact was coming from the 100° direction, the signal showed a quick small peak and was immediately followed by a valley. We hypothesize that there were two separate physical phenomena took place: (1) a quick shockwave transmitted via the helmet surface (red) and (2) a slower helmet shell bending (green) induced stretching effect. The quick shockwave transmitted via the helmet surface was perpendicular to the FBG, thus causing an expansion effect on the FBG fiber section and, thus, inducing the first peak in the spectrum. At the same time, the following helmet shell bending caused a stretching effect that is perpendicular to the FBG fiber section, which equals to a compression effect on the longitudinal direction of the FBG and, thus, induced the following valley in the spectrum. Using the same principles, all spectra generated by the impacts around the helmet can then be explained by analyzing the combined physical expansion or compression effects of each impact on the FBG fiber section.

### 3.4. Realization of True Three-Dimensional Impact Sensing

To realize sensing of the geometrically three-dimensional impact events, impact spots (from 1 to 4) with different heights were introduced upon the already applied impact positions around the helmet. The height of the impact spots is changed by adjusting the length of the rope that hangs the bowling ball. Thus, by using this approach, we introduced a third geometric parameter, latitude, to the system to realize a true three-dimensional impact condition. Figure 5 exemplifies the results with four impact directions (30°, 120°, 210°, 300°) at four impact spots. Each spectrum shown is an average of three repeated measurements (n = 3) at the highest impact energy (17.58 J). The vertical distance between each impact spot is relatively short (around 1 cm) to avoid excessive tangential impact force components. A higher spectrum similarity was observed in the FBG raw signals due highly to the short distance between spots. Specific patterns of peaks and valleys are still shown for their direct correlation with the impact directions, especially during the first 30 ms. However, the patterns of these peaks and valleys exhibit more complex signal characteristics due to the impact of different heights. Such an increase in complexity also increases the difficulty of using ML models for training and prediction. The fundamental goal of this study is to train the ML models by using complex three-dimensional datasets so that they can truly adapt to real situations and make accurate predictions.

### 3.5. Training and Performance of Selected ML Models Using the Collected FBG Datasets

As described in the experimental section, each of the ML models was first trained using 75% (selected randomly) of the database, and then the prediction performances of ML models were evaluated against the remaining 25% of the database. Figure 6 shows the best prediction results of impact magnitude (Figure 6a), direction (Figure 6b) and latitude (Figure 6c) from the test data set generated by the three ML models implemented in this study. For more specific results regarding each ML model, see Appendix A. Due to the overlap of data points, some graphs display fewer data points than others. Table 1 lists the best prediction results (indicated by the lowest CPI value) of the comprehensive evaluation, together with the six statistical parameters related to the prediction performance (for more details, please refer to Appendix A).

All ML models produced reasonably accurate predictions for impact magnitude, direction, and latitude from the collected dataset, as seen in Figure 6 and Appendix A. The statistical results in Table 1 and Appendix A proved such accuracy and ranked them based on the composite performance index (CPI). Briefly, for predictions of initial impact magnitude, the ML models can be ranked as SVM > IBK > RF > MLP-ANN, wherein the values of CPI ranged from 0.000 to 0.996. For predictions of impact direction, the ML models can be ranked as IBK > SVM > RF > MLP-ANN, wherein the values of CPI varied from 0.108 to 0.938. For predictions of impact latitude, the ML models can be ranked as IBK > SVM > RF > MLP-ANN, wherein the values of CPI varied from 0.061 to 0.926.

Here, MLP-ANN may not be a suitable model to analyze FBG sensor signals. This model performs last in the predictions of impact magnitude and latitude. Meanwhile, it consumed more time (nearly 10 times more than the SVM model) to train and analyze the data. In this study, the purpose of introducing ML for impact prediction is to assist in the fast diagnosis of potential traumatic brain injuries. Thus, the large amount of time needed for ML model analyses is not suitable (as it may result in delayed identification of the injury). The RF model did not work as effectively because the complex signals increased bias and residuals. Therefore, our study in the following steps mainly focused on the SVM and IBK models, which gave the most reliable predictions.

SVM and IBK models were able to produce predictions with higher accuracy because of the large volume of the training database, which promoted their prediction performances. Briefly, the large datasets benefit the SVM model by providing more instances for the model to develop an effective hyperplane to classify the databases in a logical manner into multiple clusters. IBK is a model which heavily depends on training instances, and large databases provide sufficient instances. Thus, the testing dataset had enough neighbor instances from the training dataset to make precise predictions. The SVM model made the best prediction, based on CPI, on impact magnitude. IBK had the best prediction results, based on CPI, regarding impact direction and latitude, but it produced less accurate predictions of impact magnitude. Note that the IBK model is based on the similarities between training instances and testing instances, which means that if the testing signals are closer to the training signals, more accurate predictions can be generated by this model.

### 3.6. Increase the Prediction Performance by Using Boosted ML Models and Modified Training Dataset

To further improve the accuracy of predictions, boosted SVM and IBK, thus far, the best two ML models were applied. Predictions of impact magnitude, direction, and latitude were produced by six derived boosted models, as selectively shown in Figure 7 and detailed in Appendix A.

Six statistical parameters pertaining to the prediction performance of the boosted model and normal ML model are enumerated in Appendix A, with the best results summarized in Table 2. According to the CPI values, S-SVM+ is the superior model for impact magnitude prediction, and S-IBK+ is the best model for impact direction and latitude predictions. Overall, boosted ML models have superior accuracy of predictions compared to their non-boosted counterparts. Besides, the smoothing filter eliminated the noise and variance from the original signal, leading to the removal of interfering factors. Meanwhile, the round method aid in the conforming of the prediction results into the training dataset formats. A combination of the smoothing filter and the round methods resulted in the optimization of predictions and the diminishment of unfavorable factors that are hard to avoid during data collection and processing.

In addition, notice that most of the feature-rich signals are located near the beginning of the impact event, as shown in Figure 3, Figure 4 and Figure 5. Therefore, it is speculated that the signal within a shorter period of time after the impact may be more representative, which is more conducive to improving the accuracy of the ML model performance. Here, the transient oscillatory signals of the first 80 ms were artificially selected and used as a new training dataset, and a preliminary test run took place using the models verified by the foregoing tests. The results are shown respectively in Appendix A and Appendix A. For impact magnitude prediction, the new training dataset improves the credibility of IBK but undermines the reliability of SVM. The new training dataset reduces some noise and interference but also loses signal features that represent the impact magnitude information. The improvement of the IBK results indicates that the first 80 ms signal of the impact magnitude has a higher spectral similarity than the later signal. For latitude prediction, the best model is IBK (80 ms), meaning the new training dataset retains the high-quality signal of the impact latitude. As we expected, S-IBK+ (80 ms) is the best-performing model for direction prediction, which means that the new training dataset and the boosted model significantly eliminate noise and bias but retain satisfactory signal details for the impact direction. To sum up, the S-SVM+ is suitable for magnitude prediction, S-IBK+ (80 ms) should be used for direction prediction, and IBK (80 ms) is a model used for latitude prediction. It should be noted that the time frame of the new training dataset (80 ms) was selected arbitrarily. In future method development, an ML algorithm instead of manual selection will be used to obtain the optimal time segment.

### 3.7. Preliminary Evaluation of the Wireless Mode Smart Helmet Sensing

To better fulfill the need for smart helmets during practical applications in sportive activities or military training, a wireless module was prepared as detailed in the Experimental section and immobilized by a tactical fabric pouch used in the modern military helmet (Appendix A). The tactical pouch provided sufficient stability and flexible cushioning during impacts, as well as effectively fixed the Redondo in place without excessive, rigid confinement. A set of preliminary test results are shown in Figure 8. 

A set of preliminary tests was conducted with nine different impact directions and five different impact magnitudes (Figure 8). Latitude variation was not introduced to simplify the process. Again, distinguishable spectral characteristics of peaks and valleys were repeatedly shown with different amplitudes. However, unlike the previous test results under the “wired” mode, these peak-and-valley signal features lost their regularity within a short period of time after the impact event. A black line was added perpendicular to the X-coordinate axis to illustrate the time point where signal regularity started to disappear. We noticed that the position of the black line might be related to the impact direction that was applied. In the 0° direction of impact, which is located farthest away from the Redondo, the signal characteristics of the first 150 ms are well maintained. However, when the impact occurred in the −30° and 30° directions, the coherence of the signals disappeared after the first 100 ms. Similarly, when the impact occurred at ±60°, the coherence of the signals disappeared at about 60–70 ms. Furthermore, the signal coherence disappeared at about 60 ms when the direction of impact was at ±90°, and the line of demarcation further moved down to 50 ms when the impact occurred in the ±120° directions. Thus, the gradual loss of coherence of the signals is presumably related to the degree of disturbance to the wireless transceiver. The larger distance of the impact from the Redondo, the smaller the influence on signals, and vice versa. The loss of signal regularity was also shown in the stack diagram that was generated by a medium-energy (10.82 J) impact (Appendix A). Typical patterns of peaks and valleys are still visible (black arrows) within the initial 50 ms (red box and the expanded figure to the right), but the “tide-like” trend found in Figure 4a can no longer be recognized.

Based on previous results, two normal ML models (SVM and IBK) and six boosted ML models (S-SVM, S-IBK, SVM+, IBK+, S-SVM+, S-IBK+) were selected for the wireless impact sensing analysis. Still, 75% of the data records were randomly selected as the training dataset, and the remaining 25% was used to test and evaluate the performance of the models. The best predictions of impact magnitudes and directions are shown in Figure 9 and Table 3. More details regarding the performances of each model can be found in Appendix A and Appendix A. All models produced reasonably accurate predictions for impact magnitudes and directions, and most predicted results fell within the 10% error range. The best model for the prediction of impact magnitudes was S-SVM+, which had the lowest CPI value. This means that the boosted model eliminated some noise from the original database and reduced the bias and variance in the predicted results. Such a result, again, demonstrated that the SVM model and its boosted version are the most suitable models for the prediction of impact magnitude. The best models for impact direction prediction were S-IBK+ and IBK+, both of which got the lowest CPI value. The same CPI value indicates that noise had few effects on the impact direction prediction. The IBK and IBK-boosted models are thus the most suitable models for impact direction prediction.

## 4. Conclusions

In this work, a single FBG-embedded smart helmet prototype with both wired and wireless modes was developed as a potential neuropathological tool for immediate sensing of blunt-force impact events. Impact magnitudes, directions, and latitudes were found to be uniquely correlated with distinctive “fingerprint” patterns of peaks and valleys that appeared in the transient oscillatory signals. These fingerprint characteristics can reveal the physical deformation parameters of the helmet and thus infer the impact details, which serve well as the basis of the ML model training and prediction. Standalone and combined ML models were employed for accurate prediction of the blunt force impact events, and the boosted models, such as S-SVM+ and S-IBK+, were found to have the optimal capability in handling such tasks. Future works will focus on the investigation of the transfer function between the blunt force impact on the helmet and the head. In addition, it is possible to monitor both blunt- and blast-force impact events using FBGs that are integrated into a flexional smart helmet.

## Figures and Tables

**Figure 1 biosensors-12-01159-f001:**
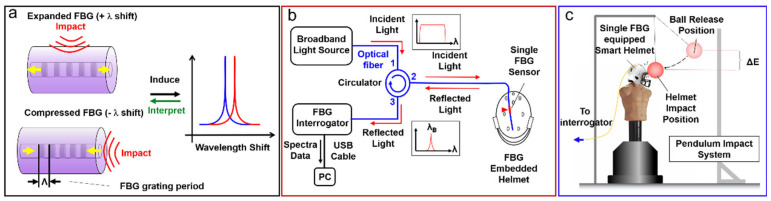
Schematics of the FBG sensors (**a**), the pendulum impact system (**b**) and the light pathway, as well as the signal interrogation method (**c**) used in this study.

**Figure 2 biosensors-12-01159-f002:**
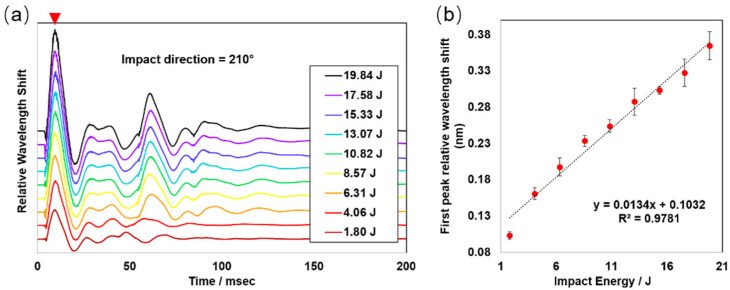
The exemplary transient oscillatory signal generated at an impact direction of 210° and the linear correlation between the impact energy and the FBG wavelength shift. (**a**) Display of transient oscillatory signals generated at an impact direction of 210° under nine different impact kinetic energy levels ranging from 1.80 J to 19.84 J. (**b**) Linear correlation between the relative wavelength shift of the first peak (red arrow) and the applied impact energy. Data points shown in the plot are averaged values ± standard deviations. All experiments were repeated three separate times under the same condition (n = 3).

**Figure 3 biosensors-12-01159-f003:**
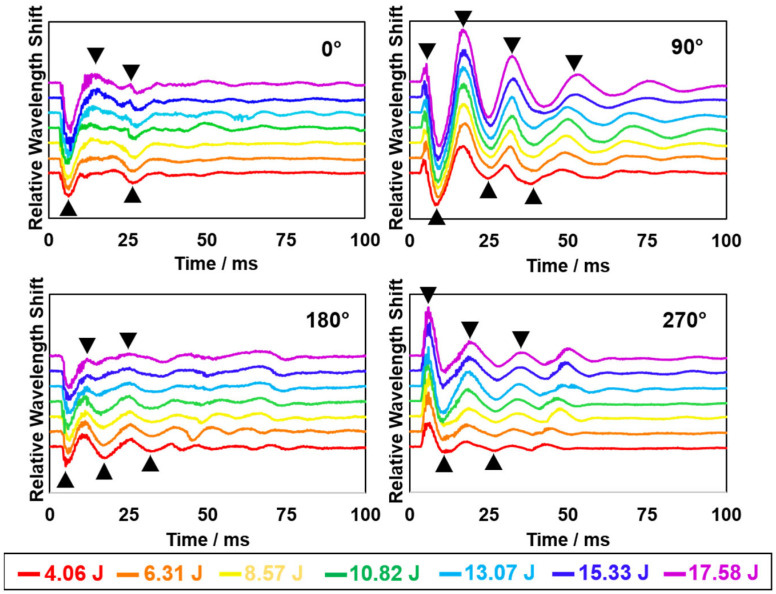
First 100 ms of the transient oscillatory signals resulting from pendulum impacts for four selected directions (0°, 90°, 180° and 270°) and seven different kinetic energy levels from 4.06 J to 17.58 J). Black arrows indicate typical spectral peaks and valleys that may serve as the “fingerprint” features for a specific impact condition.

**Figure 4 biosensors-12-01159-f004:**
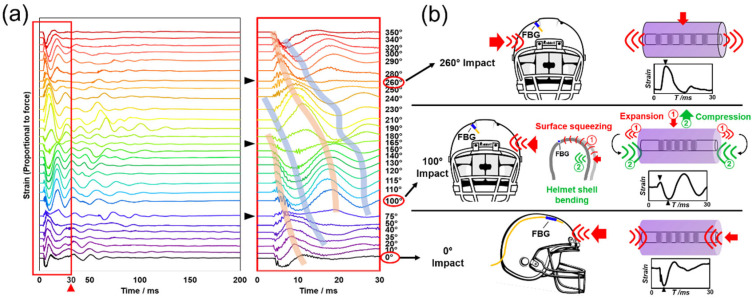
Stacked transient oscillatory signals and the physical interpretation illustrated by three selected impact directions. (**a**) A series of FBG spectra generated using 28 different impact directions and the same initial impact energy, 19.84 J. The full view of transient signals is shown on the left. The expanded view shown in the red box to the right displays the first 30 ms of data in greater detail. The black arrows indicate three peak/valley inflection points. Orange and blue bands illustrate the “tide-like” trends of the peaks and valleys. (**b**) Depictions of physical insights for three selected impact directions at 0°, 100° and 260°. Applied forces and their directions were schematically illustrated with either the front or side views of the helmet. The FBG sensor position is pinpointed by a short blue segment, and relative FBG expansions or compressions were demonstrated schematically on the right.

**Figure 5 biosensors-12-01159-f005:**
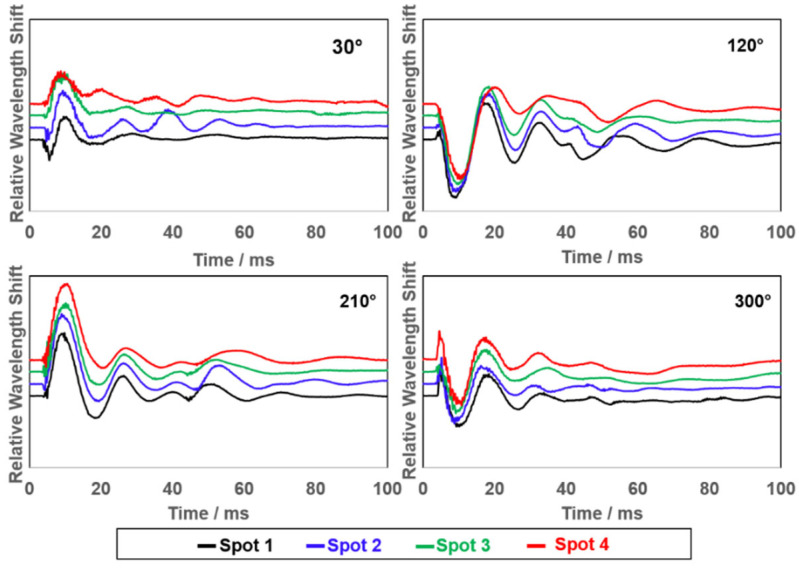
First 100 ms of the transient oscillatory signals resulting from pendulum impacts for four selected directions (30°, 120°, 210° and 300°) and four impact spots with different vertical heights from 1 to 4.

**Figure 6 biosensors-12-01159-f006:**
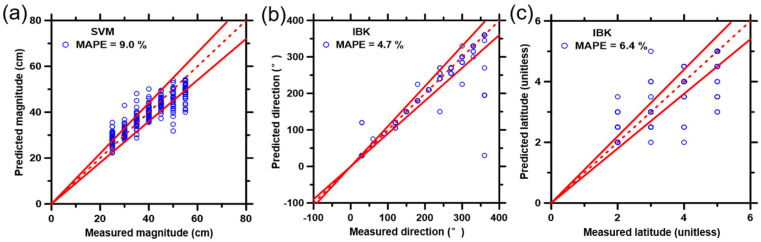
Best results of the predicted data points (25% database) versus the measured data points (75% database) on the (**a**) impact magnitude, (**b**) direction and (**c**) latitude, using four ML models that include: (1) support vector machine (SVM), (2) multilayer perceptron—artificial neural network (MLP-ANN), (3) random forest (RF), and (4) IBK. The plotted data represent 25% of the parent database that was not previously included in the training process of the ML models. The dashed line represents the line of ideality, and the solid lines represent the ±10% boundaries.

**Figure 7 biosensors-12-01159-f007:**
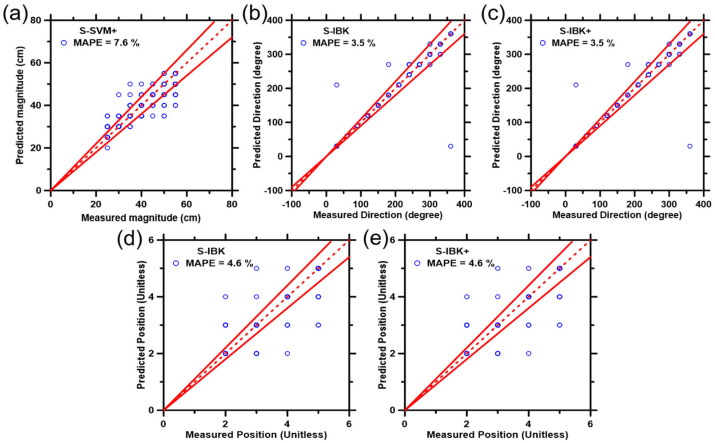
Best results of the predicted data points (25% database) versus the measured data points (75% database) on the (**a**) impact magnitude, (**b**,**c**) direction, (**d**,**e**) latitude, using the boosted ML models that include: (1) SVM+, (2) S-SVM, (3) S-SVM+, (4) IBK+, (5) S-IBK, and (6) S-IBK+. The plotted data represent 25% of the parent database that was not previously included in the training process of the ML models. The dashed line represents the line of ideality, and the solid lines represent ±10% boundaries.

**Figure 8 biosensors-12-01159-f008:**
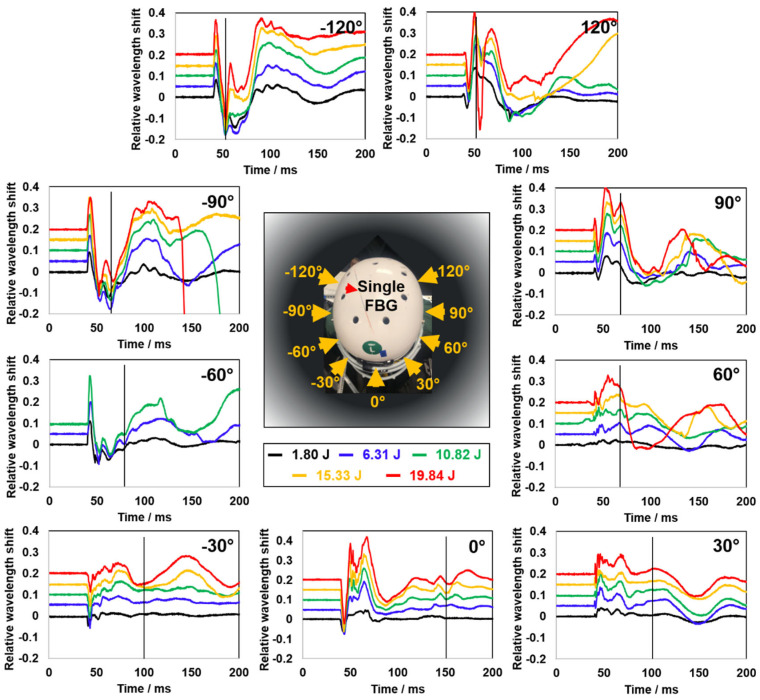
First 200 ms of the transient oscillatory signals resulting from pendulum impacts on the wireless smart helmet using nine different directions (−120°, −90°, −60°, −30°, 0°, 30°, 60°, 90°, and 120°) and five different kinetic energy levels (1.80 J, 6.31 J, 10.82 J, 15.33 J and 19.84 J). All corresponding transients were synchronized. The black line in each plot indicates the estimated time point from which the regularity of the signal started to dissipate. Yellow arrows and labels in the central photograph indicate the applied impact directions.

**Figure 9 biosensors-12-01159-f009:**
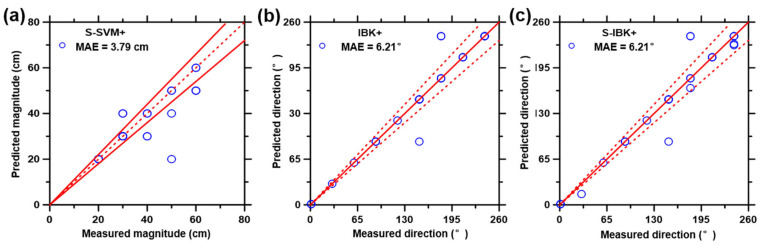
Best results of the predicted data points (25% database) versus the measured data points (75% database) on impact magnitude (**a**) and impact direction (**b**,**c**) under the wireless sensing mode. The plotted data represent 25% of the parent database that was not previously included in the training process of the ML models. The dashed line represents the line of ideality, and the solid lines represent ±10% boundaries.

**Table 1 biosensors-12-01159-t001:** Selected best prediction performance ML models on impact magnitude, direction and latitude, respectively. Five statistical parameters (i.e., R, R^2^, MAE, MAPE, and RMSE) and the composite performance index (CPI) are shown.

ML Model	R	R^2^	MAE	MAPE	RMSE	CPI
	Unitless	Unitless	cm	%	cm	Unitless
SVM-Magnitude	0.891	0.793	3.492	8.991	4.73	**0**
IBK-Direction	0.952	0.905	7.492	4.67	32.586	**0.108**
IBK-Latitude	0.915	0.837	0.204	6.373	0.454	**0.061**

**Table 2 biosensors-12-01159-t002:** Selected best prediction performance with boosted ML models on impact magnitude, direction and latitude, respectively. Five statistical parameters (i.e., R, R^2^, MAE, MAPE, and RMSE) and the composite performance index (CPI) are shown.

ML Model	R	R^2^	MAE	MAPE	RMSE	CPI
	Unitless	Unitless	cm	%	cm	Unitless
S-SVM+-Magnitude	0.889	0.790	2.980	7.635	4.680	**0.042**
S-IBK-Direction	0.971	0.942	3.960	3.536	25.385	**0.000**
S-IBK+-Direction	0.971	0.942	3.960	3.536	25.385	**0.000**
S-IBK-Latitude	0.927	0.859	0.140	4.600	0.424	**0.270**
S-IBK+-Latitude	0.927	0.859	0.140	4.600	0.424	**0.270**

**Table 3 biosensors-12-01159-t003:** Selected best prediction performance on impact magnitude, direction and latitude for the wireless sensing datasets. Five statistical parameters (i.e., R, R^2^, MAE, MAPE, and RMSE) and the composite performance index (CPI) are shown.

ML Model	R	R^2^	MAE	MAPE	RMSE	CPI
	Unitless	Unitless	cm	%	cm	Unitless
S-SVM+-Magnitude	0.852	0.726	3.793	8.506	7.656	**0.071**
IBK+-Direction	0.973	0.946	6.207	3.678	19.298	**0.340**
S-IBK+-Direction	0.973	0.946	6.207	3.678	19.298	**0.340**

## Data Availability

Not applicable.

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
