# Peer review of "A Fiber-Optic Sensor-Embedded and Machine Learning Assisted Smart Helmet for Multi-Variable Blunt Force Impact Sensing in Real Time"

_biosensors, 2022, doi:10.3390/bios12121159_

Round 1

Reviewer 1 Report

In this work, a single FBG-embedded smart helmet prototype with both wired and wireless modes was developed as a potential neuropathological tool for immediate sensing of blunt-force impact events by authors. to my view, some nice results are presented within in this papers. there is good explanation about FBG sensors and performance. But the thing I was missing in this paper, authors did not discussed other type of sensors and did not compare the performance of their proposed sensors with these sensors.

some example, you can find them below:

https://opg.optica.org/ol/abstract.cfm?uri=ol-31-9-1319

https://opg.optica.org/ol/abstract.cfm?uri=ol-42-3-486

https://www.intechopen.com/chapters/46318

https://onlinelibrary.wiley.com/doi/abs/10.1002/adom.201400460

There are other more type of optical sensors, then I really recommend that authors mentioned these papers and cite these papers.

I would like to see the revised version of this paper after major revision.

Reviewer 2 Report

 The purpose of this article is to develope a smart helmet with a single embedded fiber Bragg grating (FBG) sensor and to monitor complex blunt force impact events in real time under both wired and wireless modes. A lot of experimental results are used as datase of  machine learning algorithms to analyze the magnitude, direction and latitude of the blunt force impace.

The research seems effective to support the purpose but still have some problems:

1. There is an incorrection in this sentence: which can process about 100,000 Bril-louin et al. gained spectra along 40 km at 250 M sample/s sampling rate within 16 s.

2. The FBG sensor is embeded in the outside of helmet, which is a good buffer to protect the head, So, the blunt force applied on the helmet is far less than that on head.  The relations between the force impace to the helmet and head are not given, so it is impossible to evalue the unknown transient impact of head by analyze the hits on the helmet, and the accuracy is even far from the real situation.

3. In the abstract, the initial impact conditions with higher accuracy,is not supported by any data in the whole article. 

4. More research is needed to discover the relation between helmet and head when they are hit by blunt force, sensed by FBGs to provide best guidance for clinical practice for the on-site diagnosis of mild traumatic brain injury (mTBI).

Reviewer 3 Report

The article deals with the use of Bragg grating sensors to monitor the shocks and vibrations experienced by an athlete or a soldier in action. The article may be of interest to readers. Nevertheless, some fundamental issues need to be addressed. Indeed the choice of Bragg gratings is questionable insofar as the sensitivity is restricted to 1cm2 around the sensor. The authors deploy an arsenal of ML approaches to circumvent the problem, but which will always be present. One of the a priori solutions would be to multiplex the Bragg gratings at the headset level, which will lead to a tenfold increase in the quantity of results and the digital processing involved. We advise the authors to include at least in the discussions of the results the possibility of working on a single fiber, for example a serpentin-placed optical fiber deployed over the entire surface of the helmet and thus using a Rayleigh or Raman sensor... Which would make it possible to get significantly less dense results than in the proposed research. Ideas for improving this article could be suggested by the following 2 references, which we recommend reading and possibly adding to the list of references.

1. Drissi-Habti, M.; Raman, V.; Khadour, A.; Timorian, S. Fiber Optic Sensor Embedment Study for Multi-Parameter Strain Sensing. Sensors 201717, 667. https://doi.org/10.3390/s17040667

2. Raman, V.; Drissi-Habti, M.; Limje, P.; Khadour, A. Finer SHM-Coverage of Inter-Plies and Bondings in Smart Composite by Dual Sinusoidal Placed Distributed Optical Fiber Sensors. Sensors 201919, 742. https://doi.org/10.3390/s19030742

Round 2

Reviewer 1 Report

I can support for the publication in present form.

Reviewer 2 Report

I can accept the author's response. And the revised version is ok to be published. I wish to read further research of their research.

Reviewer 3 Report

The paper is fine